# Behavioral, Neural, and Molecular Mechanisms of Conditioned Mate Preference: The Role of Opioids and First Experiences of Sexual Reward

**DOI:** 10.3390/ijms23168928

**Published:** 2022-08-10

**Authors:** Gonzalo R. Quintana, Conall E. Mac Cionnaith, James G. Pfaus

**Affiliations:** 1Departamento de Psicología y Filosofía, Facultad de Ciencias Sociales y Jurídicas, Universidad de Tarapacá, Arica 1000007, Chile; 2Center for Studies in Behavioral Neurobiology, Department of Psychology, Concordia University, Montreal, QC H4B1R6, Canada; 3Department of Psychology and Life Sciences, Faculty of Humanities, Charles University, 182 00 Prague, Czech Republic; 4Division of Sexual Neuroscience, Center for Sexual Health and Intervention, Czech National Institute of Mental Health, 250 67 Klecany, Czech Republic

**Keywords:** opioids, dopamine, oxytocin, vasopressin, reward, first sexual experiences, mate preference, conditioned partner preference, conditioned place preference, paraphilias

## Abstract

Although mechanisms of mate preference are thought to be relatively hard-wired, experience with appetitive and consummatory sexual reward has been shown to condition preferences for partner related cues and even objects that predict sexual reward. Here, we reviewed evidence from laboratory species and humans on sexually conditioned place, partner, and ejaculatory preferences in males and females, as well as the neurochemical, molecular, and epigenetic mechanisms putatively responsible. From a comprehensive review of the available data, we concluded that opioid transmission at μ opioid receptors forms the basis of sexual pleasure and reward, which then sensitizes dopamine, oxytocin, and vasopressin systems responsible for attention, arousal, and bonding, leading to cortical activation that creates awareness of attraction and desire. First experiences with sexual reward states follow a pattern of sexual imprinting, during which partner- and/or object-related cues become crystallized by conditioning into idiosyncratic “types” that are found sexually attractive and arousing. These mechanisms tie reward and reproduction together, blending proximate and ultimate causality in the maintenance of variability within a species.

## 1. Sexual Excitation and Inhibition

Most theoretical models of sexual behavior in humans and animals describe dichotomous phases of behavior (e.g., appetitive vs. consummatory), or internal influences on behavior (e.g., excitation vs. inhibition) that flow in time and within different types of brain, spinal, and/or peripheral systems [1,2,3,4]. Orthogonal to this are linear, or multi-stage, models of sexual response. Physiological sexual responses, at least in humans, are thought to follow a four-stage model which was first conceived of by Moll [5] and made popular by Masters and Johnson [6] as their “EPOR” model of sexual Excitement (arousal and desire), genital and extragenital stimulation during the Plateau, the peak of excitement and pleasure at Orgasm, and the inhibition of excitement that occurs during Resolution (refractoriness, see Figure 1). All phases of the sexual response have both excitatory and inhibitory influences, giving behavior a substantial flexibility. Indeed, although sexual behavior is controlled by hormonal and neurochemical actions in the brain, sexual experience induces an even greater degree of plasticity that allows animals to form instrumental and Pavlovian associations that predict sexual outcomes, thereby directing the strength of sexual response. First, humans and animals must be able to respond to hormonal and neurochemical changes that signal sexual desire and arousal, and distinguish it from other sympathetic activation, such as anxiety. This ability underlies the moment-to-moment level of sexual arousability (as conceived by Whalen [7]) and defines a large part of the internal state that is commonly referred to as “sex drive”. Second, humans and animals must be able to make sense of external cues that signal sexual attraction and receptivity in potential sex partners (e.g., [8]). This ability requires a complex mix of instinct, learning, and feedback that follows a neural organization for incentive-based motivation and expectancy [9,10,11,12]. Humans and animals must be able to identify external stimuli that predict where potential sex partners can be found, to seek out, solicit, court, or otherwise work to obtain sex partners, distinguish external cues and behavioral patterns of potential sex partners from those that are not sexually receptive, and to pursue sex partners once sexual contact has been made [13]. In this way, humans and animals move in time from distal to proximal to interactive, with an ever-increasing load in sensory processing and motor sophistication as we interact more closely with either a stationary or moving and sentient array of sexual stimuli. Additionally, superimposed on this is the determination of attractivity, represented by hierarchies of stimuli from each sensory domain that are attractive to each individual.

It is becoming increasingly clear that there is a critical period of sexual behavior development that forms around an individual’s first pleasurable experiences with sexual arousal and desire, masturbation, sexual intercourse, and especially orgasm. During this period, the sensory and motor mechanics of the behavior become integrated and crystallized along with the development of preferences for ideal activities and physical features of a partner, and even objects associated with the sexual reward state. Such preferences often violate societal “norms” (e.g., as in the development of fetishes or paraphilias) and so-called “evolutionary laws” regarding features that represent genetic and reproductive strength, appearing more to be based on an egocentric evaluation of salient reward- or pleasure-related characteristics that differ from one individual to another, and create a “type” that each individual finds attractive and arousing. The formation of such experience-based preferences can be found in historical texts, such as Stendhal’s “Principle” in his work *De L’Amour* [15], in the case histories of Krafft-Ebing’s *Psychopathia Sexualis* [16], and more theoretically in the “love maps” or sexual “gestalts” proposed by Money [17]. This experiential critical period may well build upon the foundation laid by previous critical periods, especially those associated with attraction to other- or same-sex individuals, which themselves may build on the foundation laid by a critical period for gender typical vs. atypical behavior, and the sense of self as “female” or “male” [18,19].

Beyond the priming role of steroid hormones to allow animals to respond to sexual incentives (for a review see [20,21]), the orchestration of sexual behavior and partner or mate preference for all animals is a fine-tuning process between mechanisms of sexual excitation and inhibition in the central and peripherical nervous system (see Figure 2). The main neurotransmitters involved in sexual excitation are dopamine (DA), norepinephrine (NE), melanocortin (MC), and oxytocin (OXT) acting mainly in hypothalamic, limbic, and cortical regions that integrate sexual arousal, attention, and motivated behavior. Neurotransmitters involved in sexual inhibition are mainly serotonin (5-HT), and the endogenous opioids (e.g., β-endorphin) that regulate reward, satiety, sedation, and refractoriness [3,22]. For obvious ethical reasons, much of our knowledge regarding the effects of first experiences with sexual reward, the neuropharmacological and molecular mechanisms that underlie them comes from animal models (e.g., rats, quail, voles). Despite species’ differences in copulatory behavior, it is very likely that similar, if not identical, mechanisms underlie those experiences in humans [19,23].

Mesolimbic, nigrostriatal, and hypothalamic DA facilitate general attention to incentive stimuli, copulatory proficiency, and genital reflexes [3,26]. Systemic administration of DA agonists facilitates male sexual behavior [27], induces copulation in sexually sluggish [28] or sexually exhausted males [29], whereas DA antagonists impair sexual behavior in male rats [30], and eliminate sexual solicitations in females [31]. The NE system has been shown to play a role in general arousal and the control of autonomic outflow, shown to facilitate as well as inhibit male sexual behavior depending on which receptor they may bind [3,32]. For instance, yohimbine, an adrenergic receptor antagonist, enhances mounting in male rats [33], whereas lesions of noradrenergic cell bodies in the locus ceruleus increases the post-ejaculatory refractory period [34]. Finally, OXT has been implicated in sexual arousal, orgasm, satiety, and partner preference [35]. OXT cell bodies are located mainly in the paraventricular and supraoptic nuclei of the hypothalamus. Infusions of OXT into the paraventricular nucleus of male rats stimulated penile erection, while systemic administration facilitated ejaculation in male rats treated with the selective serotonin reuptake inhibitor (SSRI) fluoxetine [36,37].

The inhibitory neurotransmitters serotonin and the endogenous opioids subserve different aspects of behavioral inhibition. Serotonin neurons originate in the Raphé nuclei and send projections toward brain areas located in the brainstem, midbrain, and forebrain, as well as several structures in the periphery [3,32]. Serotonin turnover in the brain regulates satiety through a complex mechanism at many levels of processing, including the hypothalamus and prefrontal cortex, where serotonin mediates the behavioral inhibition indicative of “executive function”. Blocking serotonin reuptake with SSRIs such as fluoxetine or paroxetine inhibits sexual behavior in both female and male rats [36], whereas inhibiting its synthesis, release, or receptor binding, facilitates male sexual behavior [3,38]. Moreover, systemic injections of a serotonin synthesis inhibitor restored sexual behaviors in sexually sluggish, as well as in gonadally intact or castrated male rats [39].

Like the effects of exogenous opiates, endogenous opioids are well known for being involved in the rewarding aspects of a variety of motivated behaviors including sex [40], but also as a fundamental substrate of sexual refractoriness in humans and rats [41]. Accordingly, copulation to ejaculation increased whole brain β-endorphin content [42], and μ-opioid receptor activation in the medial preoptic area (mPOA; [43]) and ventral tegmental area (VTA; [44]), regions implicated in the control of reward in general and sexual behavior in particular [32]. Moreover, opioid use dramatically reduced sexual arousal, increased the ejaculation latency, and/or inhibited sexual responding altogether in humans and animals [22,45]. Bilateral infusions of β-endorphin to the mPOA mimicked the post-ejaculatory state behaviorally [46]. Interestingly, Chessick [47] likened the injection of synthetic opiates such as heroin to a “pharmacogenic orgasm”.

## 2. Learning and Sexual Experience

Almost every behavior is the product of the interaction between the animal’s central nervous system and its learned experiences. Yet, in order to determine which prevails over the other when it comes to partner preference, or the classical question “nature vs. nurture”, studies have shown that an animal’s own experience (proximate causality) appears to override biological predetermination (ultimate causality). Previously, it was shown that inbred mice prefer to copulate with partners of a different haplotype (or genetic parents; [48]). To evaluate whether this is a natural or learned preference, Yamazaki and collaborators took a litter of newborn mice and transferred them to foster parents whose litters were removed approximately at the same time after birth. At day 21, mice were separated into different cages with other mice of the same genotype and fostering conditions until they reached sexual maturity. At the preference test, males were given the choice to copulate with two females, one from their fostering genetic profile and another of the same H-2 haplotype. Contrary to what was previously shown [48], male mice nursed by fostered mothers chose to copulate preferentially with sexual partners that would resemble their foster mother rather than their biological mother, thus showing that H-2 selective preference is acquired by family imprinting. However, a crucial experiment showed that sexually naïve animals would choose a mate that better resembles an adoptive mother rather than the genetic mother [49]. In their experiment, Kendrick and colleagues separated male and female sheep and goats and cross-fostered them. These animals were also allowed to engage in social contact with members of their genetic species during development. When animals reached adulthood, animals were tested for social and mate preference between members of their own and foster species. Results showed that both cross-fostered males and females significantly chose more frequently to socialize and selectively mate with partners of their maternal species. These effects were more pronounced and long-lasting in males than in females. In contrast, all control animals preferred to socialize and mate exclusively with members of their own genetic species [49]. These findings provided insightful evidence on how the environment and some learning experiences shape our partner preference choices even beyond what is believed to be pre-set biologically, and suggests that epigenetic changes in attention and bonding mechanisms are altered by the experience of reward.

### 2.1. Classical Conditioning of Sexual Behavior

One of the fundamental forms of learning by which animals can acquire references from their environments and experiences was described by Pavlov [25], often referred to as Pavlovian or classical conditioning. This form of learning states that neutral cues, or conditioned stimuli (CSs), acquire meaning by their predictive association with biologically relevant cues, or unconditional stimuli (USs). This would lead ultimately to a CS becoming a priming cue that elicits a conditioned response (CR) similar to that elicited by the US, or unconditioned response (UR; e.g., [50]).

One of the first demonstrations that copulation induced a sexual reward state came from Domjan and Hall [51] who showed that male quail would remain around the vicinity of a test box window where they had previously seen a female quail. This behavior developed only if the male quail had copulated previously with that female. Subsequently, Ågmo and Berenfeld [52] found a similar phenomenon in male rats using the conditioned place preference (CPP) paradigm. In their study, male rats were placed in a CPP box with three compartments (see Figure 3). Initially, males were placed in the middle compartment and allowed to roam freely among the three compartments, each of them with different characteristics (i.e., floor texture, background color, and illumination as either dark or light). The time spent in each determined the rat’s natural preference, where males expressed a typical preference for the dark side over the light side. Male rats were then given several experiential training trials. One group of males was allowed to copulate to ejaculation with receptive females in a different chamber and immediately afterward put in the non-preferred (typically light) side of the box and left for 30 min. Two more groups were given the same sexual experience, but injected with either the opioid receptor antagonist naloxone (NAL), or the dopamine receptor antagonist pimozide (PIM) five min before copulation. Experiential control groups were injected with either NAL, PIM, or the µ-opioid agonist morphine without subsequent copulation. Finally, male rats were tested for CPP by measuring the time spent in either compartment of the box after being put in the middle compartment. Males who were allowed to copulate until ejaculation, identically to those injected with morphine, shifted their preference and spent more time in the previously non-preferred side, whereas NAL-treated males that copulated did not prefer either of the sides. Furthermore, PIM-treated males did not show a preference for either side of the box, whereas PIM-treated males who were also allowed to copulate until ejaculation did show a preference for their non-preferred side. Control groups that did not copulate or were only injected with vehicle before copulation did not show any preference for either of the sides of the box [52]. Together, these results show that the aftermath of ejaculation, while the male is in a post-ejaculatory refractory state, has rewarding properties capable of establishing positive conditioned associations with cues of the immediate context or environment, and that these associations are founded in the action of opioid, but not dopamine, neurotransmission.

As with CPP, male rats develop a conditioned ejaculatory preference (CEP) toward females bearing an odor that has been previously associated with sexual reward [55]. Working on the findings of a previous study by Graham and Desjardin [56] showing that plasma luteinizing hormone and testosterone were elevated significantly in male rats by a neutral wintergreen odor paired with copulation to ejaculation, Kippin and colleagues trained male rats to associate a neutral almond odor or no odor with copulation to ejaculation. The following three groups were tested: a paired group trained to differentiate almond-scented receptive females from unscented non-receptive females; an unpaired group trained to differentiate unscented receptive females from scented non-receptive females; and a random-paired group randomly paired with receptive and non-receptive scented females (making the odor irrelevant). In a final preference test, males were allowed to copulate freely with two receptive females, one scented and the other unscented (see Figure 3). Males in the paired group displayed a CEP for the females bearing the odor whereas males in the unpaired group displayed a CEP for the unscented females. Males in the random-paired group did not display a preference for one female over the other [55]. Subsequently, Kippin and Pfaus [57] found that the CEP depended on the male being in the presence of the scented female during the post-ejaculatory reward state. Male rats even learned to develop a preference for an unconditionally aversive odor (cadaverine) if it was associated with the postejaculatory reward state [19]. Finally, a generalized reward state can alter CEP. Ménard et al. [58] found that pairing a lemon odor in the bedding while applying gentle strokes on rat pups after separation from their mother would imprint a preference towards this odor, leading the male pups as adults to ejaculate preferentially with a lemon-scented female partner in their first sexual experience, compared to pups who did not have the lemon odor while stoked. This not only suggests that early experiences play a role in partner choice, but raises the question of whether first experiences with sexual reward in particular may shape the individual’s future partner choice by similar mechanisms.

Although rats rely heavily on their sense of smell, similar finding were reported for somatosensory and visual CSs. Domjan, Huber-McDonald and Holloway [59] used Japanese quail males and assigned them to two groups. Both were presented with an inanimate taxidermic female quail with which they could copulate for 30 sec followed by access to a sexually receptive quail hen. In the fading group, the taxidermic object was gradually covered with terrycloth over successive trials, until fully covered leaving no quail features in the last trial of training. The non-fading group was always presented with the fully covered inanimate object. After the training, each subject was tested in the training boxes for five min. with the fully covered inanimate object, except that no live female quail was introduced. Overall, males trained in the fading group spent more time around the object and displayed more copulatory behaviors (i.e., grabs, mounts, and cloacal contacts) towards the fully covered inanimate object than the males trained in the non-fading group. These data demonstrated that sexual behavior is able to be conditioned towards an inanimate stimulus object that has no natural connection with sexual reward [59]. Similarly, Köksal et al. [60] demonstrated persistence in copulation with an inanimate object after an extinction procedure only when the trained CS was the same terrycloth used by Domjan et al. [59], but not when it was a light. Male rats were also trained to show CEP for the strain of female (e.g., pigmented vs. albino) that was paired with the post-ejaculatory reward state [61]. Although the two strains clearly differ in terms of visual cues, differences in terms of accessory or main olfactory cue strength cannot be ruled out. However, male rats have also been conditioned to show a CEP for females wearing a rodent tethering jacket versus no jacket [62], suggesting a high degree of flexibility in the sensory modality of the CS that is paired with the ejaculatory reward state US. As with olfactory conditioning, the development of this CEP was blocked by systemic naloxone administration [63]. Interestingly, the same rodent jacket somatosensory CS can also be conditioned to modulate sexual arousal. Pfaus, Erikson and Talianakis [64] trained males to have their first 10 sexual experiences to ejaculation with or without wearing a rodent tethering jacket. On the final test, rats in both groups were randomly assigned to have the jacket on or jacket off. Males trained and tested with the jacket on copulated normally, as did males trained without the jacket and tested with it. However, significantly fewer males trained with the jacket copulated to ejaculation with the jacket off; and those that did displayed significantly fewer anticipatory behavior, ejaculations, and longer intromission latencies than males in the other groups. A second experiment showed that the jacket could acquire inhibitory properties if it was on the male when he was paired with a sexually non-receptive female. These data show that a somatosensory CS that the male comes into physical contact with, much like the terrycloth inanimate object used by Köksal et al. [60], can come to acquire sexually arousing properties if paired with the post-ejaculatory reward state.

Both CPP and conditioned partner preference were also demonstrated in female rats. Previous studies demonstrated that females develop a preference for the non-preferred side of the CPP box if the contextual cues are paired with paced, versus non-paced copulation in unilevel pacing chambers [54,65]. Bilevel or unilevel pacing chambers allow females to control the initiation and rate of copulation, either by running from level to level [66] or to exit from and return to the side with the male through a divider with holes that accommodate the female to pass through, but that are too small for the male to pass through [67], respectively (see Figure 3). Importantly, pacing conditions in the unilevel chamber can be modulated simply by removing the divider. With the divider, the female controls the initiation and rate of copulation. Without the divider, the male is more likely to control the initiation and rate. Females also developed a conditioned partner preference based on a neutral almond odor placed on a male rat in either bilevel chambers or unilevel chambers with the divider [68]. Using a similar conditioning training procedure as Kippin et al. [55], paired females developed a preference for scented versus unscented males which they displayed on a final open-field test with two tethered males, one scented and the other unscented: Females showed more solicitations of the scented males, more high-magnitude lordosis, and chose the scented male more often to receive his ejaculations. Systemic naloxone administration blocked the development of this conditioning [69]. Similar results were obtained when the strain of male (pigmented vs. albino) was used as a composite CS [70] and when the rodent jacket was used as a CS on the male [62].

Although pacing allows the female to control the initiation and rate of copulation, in fact it allows her to receive clitoral stimulation (CLS) and vaginocervical stimulation (VCS) from male mounts with intromissions and ejaculations at a preferred interval. Artificial CLS that mimics the rate of paced copulation not only produces CPP in sexually naïve females [71,72], but it also supports the conditioning of part of the partner preference observed by Coria-Avila and colleagues. Parada et al. [73] gave sexually naïve females repeated pairings of distributed CLS in the presence of an almond-scented gauze pad versus no CLS in the presence of an unscented gauze pad. On a subsequent open field test with two tethered males, one scented and the other unscented, the females solicited more frequently and showed higher magnitude lordosis with the scented male compared to the unscented male. However, they did not choose the scented male to receive his ejaculations selectively, suggesting that VCS may contribute to the mate choice aspects of conditioning in females.

Sexual reward states that ride on high arousal are also more potent and differentiate contextual cues from partner-related cues. For example, using CPP, male rats appeared to prefer a four-hole divider over a one-hole divider in a unilevel pacing context (see Figure 3). However, for CEP to be conditioned, rats must be trained with the one-hole divider [74]. Similar results were obtained in female rats [75], especially as it concerned the female’s choice of male from whom to take ejaculations. This typically imposes a greater inter-intromission interval for the male and longer return latency of the female after mounts, intromissions, and ejaculations, especially if the male puts his head into the hole to try to reach the female. Females typically impose a longer inter-intromission interval prior to a male’s ejaculation in bilevel chambers [66]. They do this by running from level to level more times, which forces the male to chase them for longer periods. It appears that extending the time the animals have to wait to get genital stimulation increases their arousal, which, in turn, increases the potency of the postejaculatory or CLS reward state.

### 2.2. Instrumental Learning and Sexual Behavior

Instrumental learning originates with Thorndike’s law of effect [76] that states that if a behavior in the presence of a stimulus is followed by a satisfying consequence, the established association between the behavior and the stimulus becomes strengthened. Similarly, when the behavior is followed by an aversive consequence, the association weakens. This type of conditioning defines a contingency between responses and their reinforcers, and the term “operant” was coined by Skinner [77]; for behaviors that “operate” on the environment), who advanced the study of this form of learning by allowing animals to freely perform the behavior, as opposed to discrete-trial procedures. That way, animals were free to perform and repeat the operant response over and over. Thus, the behavior could be shaped based on different schedules of reinforcement or punishment [78].

Several demonstrations of male rats performing lever-pressing operants to gain access to sexually receptive females were documented [79,80], along with overcoming obstacles [81], or crossing shock grids or other aversive tasks [38,82,83,84]. Other preparations that manipulated brain neurochemistry and anatomy were also explored. For example, axon-sparing neurotoxic lesions of the basolateral amygdala disrupted lever pressing for a secondary sexual reinforcer (a stimulus light paired with access to a sexually receptive female), whereas it did not affect copulation in males [85]. In contrast, the same lesions of the mPOA disrupted copulation, leaving conditioned lever pressing virtually unaffected at the time of testing [86].

### 2.3. First Experiences of Sexual Reward

In every case, the studies reviewed above controlled an animal’s first sexual experience and provided the context for sexual reward or non-reward during that and subsequent sexual experiences. In humans, first sexual experiences may or may not coincide with sexual reward, and indeed sexual reward from orgasm or other sensory stimulation may not occur every time a person has sex. However, more controlled manipulations of first experiences have been tested experimentally. In order to elucidate what first experiences with neutral or biologically relevant stimuli actually do, researchers exposed animals to either the CS or US before training for conditioned preference, resulting in different outcomes depending on what is being pre-exposed and for how long.

*US pre-exposure (blocking)*: US pre-exposure is the phenomenon in which there is retardation or outright blocking of the establishment of a CR if its CS is paired with the US in a context in which the animal was previously exposed to the US alone (e.g., [87]). For example, Taylor [88] conditioned the blinking response of human participants using an air puff signaled by a light. Before training, one group received several presentations of the air puff in three different intensities to the cornea of their eyes without light. The number of eye-blink responses was greater in the group that was not pre-exposed to the air puff, whereas in the pre-exposed group the number of eye-blink responses was in an indirect correlation with the intensity of the air puff during the pre-exposure phase. Two different explanations were proposed, one associative and the other non-associative. The former proposes that the US pre-exposure effect is due primarily to an association between the US and cues in the context in which the initial US pre-exposure takes place prior to training (e.g., [50,87]). The latter claims that by pre-exposing the US, there is a reduction in initial emotional reactivity of the animal’s response due to general habituation that reduces the salience of the US, and thereby attenuates subsequent excitatory conditioning (e.g., [89]). In a study of blocking [90], male rats were given one or five copulatory experiences to one ejaculation each with unscented receptive females. Subsequently, males were given another 10 experiences with almond-scented receptive females, and later on tested in an open field on their 11th trial with two receptive females, one scented and the other unscented. Regardless of the number of US pre-exposures to the unscented female, males did not display any CEP to the scented or unscented females, indicating that the first experience of the post-ejaculatory reward state was sufficient to block the ability of the odor to control the associative strength. Likewise, CLS is no longer capable of inducing CPP in female rats that have had experience with paced copulation in a bilevel chamber [91].

*CS pre-exposure (latent inhibition)*: As with blocking, pre-exposing an animal to the CS produces a disruption or retardation of a subsequent trained association with the same CS (e.g., [92]). For instance, animals pre-exposed to a saline solution used later on to train conditioned taste aversion showed retardation of the association in comparison to a control group that was not exposed to it previously [93]. Most theories coincide in that latent inhibition is the result of a reduction in associability or attention to CS during pre-exposure [94]. This learning phenomenon and its properties highlights the ability of animals to form new associations through passive, non-reinforced pre-exposure of CSs, demonstrating that previous experiences influence them when being trained to learn new associations with neutral cues. Zamble, Mitchell and Findlay [95] demonstrated that single CS or contextual cues can facilitate copulation in Japanese quail (i.e., reduced ejaculation latency) if they predicted copulation with a receptive female. However, when animals were pre-exposed enough times to the mating context, the background cues were shown to be subjected to latent inhibition. Similarly, Quintana et al. [96] gave male rats one or five pre-exposures to the neutral almond odor on gauze prior to giving males 10 copulatory trials to ejaculation with scented receptive females. On the final test, males were placed into an open field with two receptive females, one scented and the other unscented. The group provided with one odor pre-exposure displayed significant CEP, whereas the group that received five pre-exposures did not.

### 2.4. Potentiating or Inhibiting Conditioning during First Experiences with Reward

In addition to high arousal, the ability of certain drugs to potentiate conditioning during first experiences with sexual reward was examined. The results of these studies begin to frame the neurochemical systems that sensitize during first experiences with sexual reward. Ménard et al. [97] found that parvocellular OXT neurons in the paraventricular nucleus of the hypothalamus (PVN) that project to other brain regions and magnocellular vasopressin (AVP) neurons in the supraoptic nucleus of the hypothalamus (SON) that project to the posterior pituitary were conditionally activated by the almond odor CS in male rats. A subsequent experiment injected either OXT, AVP, or saline, subcutaneously to males prior to their first sexual experience to ejaculation with receptive females. All males were injected with saline prior to their second sexual experience in an open field with two receptive females, one scented and the other unscented. Only males injected with OXT during their first sexual experience to ejaculation showed a weak but statistically significant CEP during the open field trial.

Likewise, female rats given repeated paced copulation with an unscented male rat displayed classic mate-guarding behavior when placed into an open field with their familiar male and a competitor female [98]. This behavior consists of hovering and presenting postures close to the male, attempts to block access of the competitor female to the male by getting between them, and the aggressive mounting of, or outright fighting with the competitor female if she solicits the male. In contrast, if the conditioned female is receptive and with a novel male and a receptive competitor female, she displays a species’ typical pattern of competitive solicitations and interceptions of the male, similar to the observations of McClintock [99]. Holley et al. [100] found that conditioned females exposed to their familiar males also had significant activation of parvocellular OXT neurons in the PVN and magnocellular OXT and AVP neurons in the SON. Holley et al. [100] also examined the effect of subcutaneous OXT, AVP, or saline injections during the females’ first paced experiences with a male on subsequent mate guarding during their second experience. All females were injected with saline prior to their second sexual experience, which took place in an open field with their familiar male and a competitor female. Females injected with OXT showed significantly more incidents of hovering and presenting relative to saline controls, made significantly more solicitations, and received significantly more intromissions and ejaculations. In contrast, females injected with AVP displayed significantly more incidents of conspecific blocking of the competitor female relative to females previously treated with OXT or saline. Thus, in both male and female rats, the hyper-OXT state augmented the experience of sexual reward and produced significant learning of a partner preference. A hyper-AVP state did not do this in males, but potentiated the vigilance of the female toward the competitor. Finally, female rats that received systemic injections of naloxone during their first paced sexual experiences demonstrated a lack of sexual interest, solicitations, and lordosis on a final saline test despite being fully hormonally primed [19]. Similarly, females that received subcutaneous injections of the lysine-specific demethylase inhibitor oryzon (to prevent epigenetic alterations resulting from sexual experience) during paced conditioning trials with the same male did not display any mate-guarding behaviors with that male, nor did they display an increased activation of OXT or AVP neurons in either the PVN or SON [101]. Male rats that received systemic naloxone injections during their first sexual experiences with almond-scented females displayed a preference for the unscented female on the final test [74]. These data indicate an important interaction of the sexual reward state with mechanisms of arousal, attention, and bonding that were induced by both contextual and partner related cues that predict the reward state. How might this occur?

## 3. Brain Activation by Sexual Experience and Cues Associated with Sexual Reward

As with other motivational systems and rewards, experience with sexual reward changes the brain and sensitizes behaviors aimed at acquiring sexual reward [102,103,104,105]. These changes occur within neural excitatory systems, including up-regulation of nitric oxide synthase in the mPOA (NOS; [106,107]), OXT receptor in the VMH and central amygdala [108], immediate early gene proteins such as ΔFosB in the VTA, NAc, medial prefrontal cortex and infralimbic cortex [104], enhanced synthesis of brain-derived neurotrophic factor (BDNF) and its tyrosine kinase receptor B (trkB; [104]), sensitized DA release in the NAc and medial prefrontal cortex [109,110,111,112], and altered DA cell morphology and enhanced function in the VTA [113]. In addition to the activation of OXT and VP neurons in the PVN and SON reviewed above, cues associated with sexual reward activate overlapping neural circuits for sexual excitation, reward, and bonding [114,115]; similar to the regions activated by direct copulatory stimulation [72,116] and stimulate DA release conditionally in the NAc [19]. Pitchers et al. [113] found that the activation of ΔFosB in the NAc was necessary for the reinforcing properties of sexual reward in male rats, presumably on the role played by mesolimbic DA transmission in this critical terminal region.

## 4. Dual Role of Endogenous Opioids in Sexual Behavior

There are three main types of endogenous opioids: endorphins, enkephalins, and dynorphins. They are the result of an enzymatic process of three different precursor molecules, pro-opiomelanocortin (from which melanocortins such as α-MSH and corticotropins such as ACTH are derived), pro-enkephalin, and pro-dynorphin [117,118,119]. There are also three types of opioid receptors, µ, κ, and δ. These receptors are located predominantly in hypothalamic, limbic, and cortical areas [120,121] and have different roles in the control of sexual behavior. Although enkephalins and dynorphins are the natural ligands of δ and κ receptors, respectively, the µ receptor has a number of endogenous ligands that bind to it, including endomorphins, endorphins, enkephalins, and morphiceptin (derived from the milk protein β-casein). Upon agonist stimulation, opioid receptors are internalized into clathrin-coated endosomes and are no longer expressed on the cell surface [43,122]. Opioid receptor internalization, then, can be used as a measure of agonist binding.

### 4.1. Inhibition and Refractoriness or Disinhibition

During copulation, and after several vaginal intromissions, male rats ejaculate and fall into a period of sexual quiescence in which another erection cannot be achieved in the span of a few minutes, a period known as the post-ejaculatory interval (PEI) or refractory phase (for a review see [6,32]). Likewise, female rats have relatively short refractory periods (30–60 s) after ejaculation or distributed CLS [67], and much longer refractory periods after multiple VCSs that define estrous termination [123].

Many of these inhibitory behavioral responses can be mimicked by the systemic or central administration of opiate drugs or by the central administration of endogenous opioids [22,40]. For example, β-endorphin infused into the mPOA of sexually experienced male rats inhibited their copulatory behaviors in a dose-dependent fashion [46,124]. Furthermore, compared to sexually active males, sexually inactive males demonstrated an increment of endogenous opioid octapeptide Met-Arg^6^-Gly^7^-Leu^8^ in the hypothalamus [125], as well as an increment of pro-enkephalin and pro-dynorphin mRNA expression in the paraventricular nucleus [126]. Infusions of morphiceptin into the mPOA of male rats produced a delay in their initiation of copulation compared to the control animals infused with a vehicle solution [127]. Moreover, inhibition of copulatory behaviors (e.g., copulation latency and ejaculation) was found when infusing a κ opioid receptor agonist (U-50488H), an effect that disappeared differentially when a κ opioid receptor antagonist (nor-binaltorphamine) was infused into the VTA, mPOA, or NAc [128]. Facilitation of male sexual behavior was reported when the longer-acting opioid receptor antagonist, naltrexone, was administered in sexually inactive males [129], sexually naïve males [130], and sexually satiated males [131], and also reduced the ejaculation latency and increased ejaculation frequency in sexually active male rats [132]. However, some of these effects may be disinhibitory and counteract the effect of stress-induced opioid release (e.g., [130]) or a progressive buildup of opioid binding after multiple ejaculations [131]. These data suggest that the NAc, mPOA, and VTA are important sites for the opioid mediation of sexual inhibition.

Much of the work on opioids in females has been limited to lordosis, with selective μ receptor activation being inhibitory and selective δ or κ activation having a facilitative effect [22,133]. NAL in all cases reversed these effects. In many cases, however, endogenous β-endorphin or morphiceptin had a dose-dependent dual effect, with low doses facilitating and higher doses inhibiting lordosis. Different opioids have inhibitory effects in different brain regions. For example, endomorphins inhibit lordosis after intraventricular infusions or site-specific infusions to the lateral septum and diagonal band of Broca, whereas the synthetic μ agonist D-Ala2-Met5-enkephalin (DALA) inhibited lordosis following bilateral infusions to the mPOA and VMH [134]. Facilitation of proceptive behaviors was observed following infusions of a δ agonist to the third ventricle of female rats primed with estradiol and progesterone, but not estradiol alone [133]. Recently, Johnson, Hong and Micevych [135] inhibited lordosis using the optogenetic activation of β-endorphin terminals in the medial nucleus of the mPOA of female POMC-cre mice primed with estradiol and progesterone. The inhibition was accompanied by the internalization of μ opioid receptors.

### 4.2. Reward-Related Sensitization of Incentive Cues

Opiates and opioids are well known for their rewarding or positively reinforcing properties. Morphine and β-endorphin both induce CPP and support both peripheral and intracranial self-administration [136,137,138,139,140]. Likewise, copulation and ejaculation in males, or paced copulation/distributed CLS in females, are crucial rewarding events in the establishment of CPP [141], CEP [57,74], and conditioned partner preference [69]. As with all reward-related learning, the administration of the opioid receptor antagonist NAL or naltrexone block the acquisition of these conditioned responses indicating that endogenous opioid transmission is critical. Indeed, whole-brain β-endorphin content increases dramatically after ejaculation in male rats [42], with specific increases in enkephalin content observed in the midbrain and hypothalamus [93]. Copulation to ejaculation also induces µ-receptor internalization (a marker of ligand-induced receptor activity) in the mPOA [43] and VTA [44]. More specifically, it has been shown that different numbers of ejaculations render different µ- and δ-receptor internalization in the VTA, and although the internalization of both receptors increased as a result of ejaculation, only µ-receptor internalization was correlated with the number of times a male ejaculated [41,142], suggesting that this receptor carries the reward or pleasure signal. Indeed, bilateral infusions of DALA to the mPOA, induce significant CPP [141]. These data are consistent with the overall notion of μ receptors in many reward-related regions of the brain carrying the reward signal (e.g., [143]), with δ receptors in the NAc shell further enhancing reward [144]. The reward signal impinges on neural systems for motivation, or more specifically, desire and attention (see Figure 4).

### 4.3. Antagonism of Opioid Reward

As mentioned previously, copulatory-induced CPP in both females and males, CEP in males and conditioned partner preference in females can be disrupted by the administration of NAL during training. Appetitive aspects of sexual behavior that denote desire are strongly affected by the blunting of opioid reward. Male rats in bilevel chambers naturally develop a characteristic level searching behavior prior to the introduction of a sexually receptive female [146]. This behavior was blocked by the systemic administration of NAL [147]. In females, repeated training trials with NAL resulted in a dramatic decline in solicitations and lordosis overall, and an increase in the display of rejection responses made toward both males in the open field choice test, despite females being fully primed with estradiol and progesterone [69]. However, regarding CEP in males, if the training with NAL is paired exclusively with the scented female, the disruption of CEP for the scented female is typically a shift in choice to the unscented female during the final open field choice test, despite those males copulating to multiple ejaculations during the training trials. A similar phenomenon was reported in male prairie voles. Typically, males display a partner preference for a familiar female with whom they cohabitated and copulated relative to an unfamiliar female. However, if that cohabitation occurred under the influence of naltrexone, the males shift their preference to the unfamiliar female [148]. This suggests that the antagonism of opioid receptors during copulation induces a state of non-reward that is associated with the familiar female, and shifts the male’s preference to the unfamiliar or novel female. A state of non-reward induced by frustration also shifts the preference of female rats toward the male associated with it. Parada et al. [91] gave females distributed CLS in the presence of a scented but inaccessible male using a wire-mesh screen to divide the unilevel pacing chamber. On the final open field choice test, females solicited and received ejaculations selectively from the unscented male relative to the scented male.

Brain regions involved in this behaviour have begun to be examined. Burkett et al. [148] infused the selective μ receptor antagonist D-Phe-Cys-Tyr-D-Trp-Orn-Thr-Pen-Thr-NH2 (CTOP) to either the NAc or caudate-putamen (CP) of male prairie voles (identical to the group above that received naltrexone). Infusions to the CP, but not NAc, disrupted partner preference and the time spent huddling, relative to saline infusions. Quintana et al. [149] examined the role of NAL infusions to the mPOA or VTA on CEP. Relative to males infused with saline, infusions of NAL to the mPOA shifted the preference towards the unfamiliar female (similar to peripheral injections of NAL), whereas infusions to the VTA abolished CEP completely. Taken together with the data of Ågmo and Gomez [141], the mPOA would appear to be a region where the reward value of sexual behavior (and perhaps other motivated behaviors) is determined by opioid action. In contrast, regions of the mesolimbic pathway (e.g., VTA and NAc) appear to be involved in the opioid sensitization of attention. This raises a paradox: opioids inhibit sexual behavior in these regions but, with time, sensitize the systems responsible for sexual incentive motivation, including reward-related attention and partner preference. How might this work at a neuropharmacological and molecular level?

## 5. Neuropharmacological, Signal Transduction, and Molecular Mechanisms of Opioid Sensitization

### 5.1. Mesolimbic DA Sensitization

The ability of sexual reward to induce CPP, CEP, partner preference, and to enhance appetitive sexual behaviors, must involve a process of sensitized attention to environmental and partner-related cues. One of the classic mechanisms of attention to reward-related cues is the sensitized activation of mesolimbic/mesocortical DA [150,151,152,153,154] (see Figure 5). The A10 DA neurons that make up this pathway originate in the ventromedial portion of the VTA and project to a number of cortical and limbic sites, including the prefrontal cortex, anterior cingulate, septum, amygdala, and NAc [155,156,157]. In prairie voles, DA release in the NAc is particularly important for pair-bond formation [158], with DA actions on the D2 receptor family being important for formation and actions on the D1 receptor family important for retention of the pair bond [159]. Sensitized DA cell body activation in the VTA, sensitized DA release in the septum, and/or sensitized DA release in the NAc, can be induced by opioids acting at the μ or κ receptor, respectively, and plays a major role in Pavlovian learning. This has been shown for stimulus–drug associations [160,161] and other forms of incentive learning in which an external stimulus predicts a central reward state. When conditioned, such stimuli become particularly resistant to extinction. We note that DA release in response to the familiar odor associated with CEP caused an increase in NAc DA release to an average of 180% to 200% in paired males, but produced only a small and non-significant increase in unpaired males [19]. Similarly, paired males also showed a slight but nonsignificant increase to a novel odor not associated with CEP. This indicates that the familiar odor as a predictor of sexual reward (and CEP) had sensitized DA release in the NAc. Although not tested, it is likely that NAL blocked this sensitization given that it blocked the acquisition of CEP. Indeed, NAL prevents the ability of morphine infusions to the VTA to induce CPP, and naltrexone treatment blocks the reinstatement of both heroin and cocaine self-administration in rats [162].

The mechanisms responsible for DA sensitization by μ opioid receptor agonists have been studied in the VTA and NAc. In the VTA, DA neurons are typically under tonic inhibition by GABA neurons from the ventral pallidum or rostromedial tegmental nucleus, and by local interneurons that are activated by glutamate outputs from regions such as the cortex. These interneurons are inhibited by agonist action at μ receptors, thus disinhibiting DA neuronal activity [44,166]. However, DA neurons can be activated by glutamate projections directly, and presynaptic μ receptors on glutamate terminals can inhibit this activation. This provides opioid, GABA, and glutamate projections to the VTA with the ability to inhibit or facilitate DA cell body activation. In the NAc, DA release occurs in the following two forms: transient or phasic DA release caused by the depolarization of DA cell bodies in the VTA, and sustained or tonic release regulated by afferents from the PFC and amygdala to the NAc [167]. In male rats, the appetitive period of sexual behavior is characterized by phasic DA release in the NAc, whereas copulation is characterized by tonic release until ejaculation, when DA levels plummet and remain low for the absolute refractory period, then slowly rise up during the relative refractory period [109,168]. The decrease in DA is caused in part by opioid and serotonin actions that inhibit DA cell bodies and presynaptic terminals in the NAc. Infusions of morphine or dynorphin into the VTA increases DA release in the NAc, which ultimately facilitates male sexual behavior [169]; however, μ receptor agonists increase the length of the PEI (e.g., [170]), again showing that opioids acting on μ opioid receptors can facilitate or inhibit different aspects of copulation in males. Importantly, Kippin and Pfaus [57] determined that being in the presence of the scented female during the post-ejaculatory refractory period is when the incubation of learning occurs that results in the CEP. This is a period when DA activity is inhibited, but opioid activity is enhanced. In particular, this enhancement comes from the following two opioids working simultaneously: β-endorphin activation of μ receptors on GABA interneurons in the VTA and dynorphin activation of κ receptors on presynaptic DA terminals in the NAc.

### 5.2. DA Related Signal Transduction and Genomic Mechanisms

The signal transduction and molecular mechanisms that underlie opioid sensitization of DA transmission occur during this period of DA suppression. All opioid receptors are coupled to G-proteins, making opioid receptors members of the GPCR “family”. Agonist binding is diminished by guanine nucleotides and agonist-stimulated GTPase activity. In addition, all opioid receptors inhibit adenylate cyclase [171]. Many GPCRs stimulate mitogen-activated protein kinase (MAPK or ERK) activity, although MAPK activation is not dependent on μ receptor internalization [172]. Although opioid receptors inhibit voltage-gated N-type Ca^2+^ channels, they activate inwardly rectifying K^+^ channels and phospholipase C-β (PLC-β), responsible for the splitting of membrane bound phosphatidylinositol into its constituent second messengers, inositol trisphosphate (IP3) which liberates intracellular Ca^2+^ from the endoplasmic reticulum, and diacylglycerol (DAG) which can activate Ca^2+^ binding proteins and enzymes [173]. However, in many cases, this requires the additional stimulation of Gq-coupled receptors, which also stimulate Ca^2+^ release from intracellular stores via the IP3 pathway. Ultimately, these multiplicative actions stimulate more and more intracellular Ca^2+^ that can activate binding proteins in the cell membrane (e.g., PKC and CAM kinase) and stimulate calcium response element binding (CREB) proteins to regulate gene expression. This allows opioids to regulate the transcription of proteins linked to μ receptor trafficking, such as G-protein coupled receptor kinase 2 (GRK2) and β-arrestin 2, as well as receptors for other neurotransmitters, such as DA receptors, NMDA receptors, GABA-A receptors, and alpha adrenergic 2A receptors [174]. Interestingly, β-arrestin 2 may regulate opioid sensitized DA release. Transgenic β-arrestin 2 knockout mice exhibit a greater sensitization of DA release and greater reward in response to alcohol [175], which is known to activate opioid-mediated DA sensitization (see Figure 4 and Figure 6).

### 5.3. Downstream Activation of OXT and AVP Neurons

Of the many neuropeptides that modulate sexual behavior, OXT and AVP figure prominently in sexual arousal [176,177,178,179] and the establishment of monogamous partner preferences in prairie voles [180,181,182,183,184,185,186]. As mentioned above, the mating-induced preference by male prairie voles for a familiar female can be inhibited by naltrexone, suggesting that opioids play an important role in their development. However, so does DA. Opioids and DA both participate in the sensitization and activation of OXT and AVP neurons. The following two populations of OXT neurons exist in rats and other rodents that comprise two systems within the PVN and SON of the hypothalamus [187]: small parvocellular neurons that project largely to midbrain (periaqueductal grey), brainstem, and spinal cord, and larger magnocellular neurons that project diffusely to limbic and forebrain sites, and to the posterior pituitary [188,189,190]. Parvocellular OXT neurons also make direct dendritic contact with magnocellular OXT neurons in the SON [191] and regulate OXT release from those neurons into the posterior pituitary vasculature. Thus, OXT can coordinate sexual arousal with partner preference by both central and peripheral actions.

As an antidiuretic hormone, AVP increases water retention by the kidneys. As a vasopressor, AVP causes vasoconstriction and increases blood pressure. Central activation of V1a receptors in the ventral pallidum is also critical for the display of partner preference in prairie voles, and the overexpression of V1a receptors in a promiscuous vole species can induce monogamous-like partner preferences [192]. Despite its clear role in monogamous partner preferences, injections of AVP during early experience with sexual reward did not augment CEP in males or conditioned partner preferences in females. This leaves conditional activation of OXT as a potential integrator of sexual arousal and partner preference.

The conditional activation of OXT neurons in particular by cues associated with sexual reward likely helps to assure reproductive success with preferred partners. This would occur by potentiating the coordination of sexual arousal, ejaculation, and the cervico–uterine reflex contractions that figure in sperm transport, in the presence of one’s preferred partner or perhaps even just in cues associated with sexual reward. Indeed, OXT receptors were found in the penis [193], in the glans and corpora of the clitoris and in the cervix [194]. In addition to potentiating sexual partner preference, the peripheral administration of OXT shortens the duration of female rat sexual behavior by facilitating estrous termination [194]. Females that take themselves out of the mating context for over 10 min assure paternity for the last male’s ejaculation [195]. This behaviour following the selective acceptance of ejaculations by the preferred familiar male sets up a behavioral system to assure paternity. Likewise, CEP with the same female would also be expected to facilitate paternity, especially if the male was also the female’s preferred choice. This phenomenon of OXT coordination links ultimate causality with proximate causality, making successful reproduction more likely based on a prior history of orgasm-like sexual reward with the preferred partner (see Figure 7).

### 5.4. Epigenetic Mechanisms

The interaction between the environment, behavior, genes, and the underlying mechanisms by which they influence each other is an important focus for research on sexual behavior and partner preference [197,198]. By manipulating the way DNA is unfolded from histones and expressed, either by silencing or expressing certain parts of the gene, scientists observed behavioral changes in different animal models. Epigenetics refers to the study of how environmental changes modify the way the genome is expressed through the manipulation of the enzyme responsible for the addition or removal of epigenetic tags in histone proteins, without altering the DNA sequence [199]. For instance, through the administration of a histone deacetylase inhibitor (HDACi), a drug that promotes DNA acetylation and transcriptional activation, sexually naïve female voles developed a partner preference by simply being exposed to a male vole partner in the absence of mating [200]. Conversely, demethylation also promotes the unpacking of genes [201], and the administration of oryzon, a specific inhibitor of lysine specific demethylase-1 (LSD-1), blocked the development of conditioned mate guarding behavior in the female rat in a manner similar to NAL [101]. Females in the group that received oryzon prior to each training trial copulated normally with males during the training trials indicating that sexual arousal and opioid reward were processed normally despite demethylase inhibition. However, those females failed to display the mate guarding behaviors shown by females that received saline before each trial. Moreover, double-labeled cell counts for Fos within OXT and AVP neurons were significantly lower in the oryzon-treated group compared to the saline-treated group in both the PVN and SON. Histone modifications are a key element in gene regulation through chromatin remodeling. LSD-1 demethylases have been shown to demethylate repressive histone markers thus leading to transcriptional activation [201], and blocking this keeps those genes repressed. The results of Holley et al. [101] suggest that genes involved in the activation of OXT and AVP neurons, but not the rewarding effects of opioids, are especially affected by LSD-1 demethylase inhibition. Interestingly, D2Rs and D3Rs were found on both OXT and AVP neurons in the PVN that also express BDNF [202,203]. Demethylase inhibition reduces the viable expression of a number of genes related to DA receptor function, including an attenuation of D2R-activation of a G protein-coupled inwardly rectifying potassium (GIRK) channel [204], which, in turn, would be expected to diminish the conditional activation of OXT and AVP neurons. These data also suggest that the activation of OXT and AVP neurons occurs downstream of the DA neuronal populations sensitized by opioids. Indeed, neuroanatomical and microinjection studies show that A13 incerto-hypothalamic DA neurons project axons to the PVN and are themselves activated by efferents stimulated by DA release in the NAc [196].

### 5.5. Molecular Mechanisms of Opioid Sensitization of OXT Neurons and the Role of DA

Opioids also sensitize OXT neurons directly by their actions on μ opioid receptors. This occurs by a complex interaction with the activation of Gi proteins that inhibit cAMP, but disinhibit MAP kinase signaling (via ERK, p38MAPK, and JNK pathways). In turn, MAPK pathways activate the transcription of genes, one of which is the Cluster of Differentiation 38 (CD38) gene, a cyclic ADP ribose hydrolase that acts as a marker of cell types in addition to being an activator of B cells and T cells [205]. However, its enzymatic activity also forms cyclic ADP ribose and nicotinic acid adenine dinucleotide phosphate which, similar to IP3, release Ca^2+^ from intracellular stores. This action is important for OXT neurotransmission [206,207,208,209,210,211]. Diminished CD38 action figures in autism-spectrum disorders [212,213] and CD38 knockout mice show impaired object- and social-recognition memory [208]. Thus, sexual reward and pleasure driven by μ opioid actions on OXT neuronal membranes may augment the synthesis of CD38, which in turn could sensitize OXT release in the presence of sexual reward-related cues. If true, then when DA release activates OXT neurons (via opioid sensitized mesolimbic DA transmission that activates efferents to A13 DA neurons projecting to the PVN and SON), the augmented DA activation would stimulate more OXT release, in addition to the DA-sensitized and focused appetitive responses made toward preferred sex partners or merely stimuli associated with sexual reward (see Figure 8).

## 6. First Experiences of Sexual Reward in Humans; or the Path of Cupid’s Arrow

First sexual experiences in humans typically involve masturbation to orgasm with the use of sexual content material (e.g., pornography) before having first copulatory contact with another person [214,215]. Although first sexual experiences likely involve arousal and desire, they do not necessarily end in orgasm. This is especially true for women [216,217,218,219]. It is also the case that not all orgasms for men or women are alike, and will depend, as with our rats and their orgasm-like responses [220], on the context. For example, orgasms experienced in the context of coercive sex, compliant sex, and/or pressure to have an orgasm, not only dampen the pleasure of orgasm, but make it feel “bad” (e.g., [221]). Likewise, orgasms can be diminished in quality when sex is frustrating, inhibited, habitual, or otherwise during low arousal [19,222,223,224,225,226] or when heteronormative scripts that emphasize male pleasure from vaginal penetration alone are driving the sexual interaction are followed [227,228] (see also [229]). Even ejaculation quality in terms of sperm count is diminished when sexual arousal and orgasm quality are diminished [230]. Additionally, of course, orgasm quality can be diminished by relationship stress, anhedonia, and depression [231,232,233,234], and conversely with the use of selective serotonin reuptake inhibitors (SSRIs) used to treat depression [235,236,237,238,239,240,241,242]. This is not to say that sex cannot be pleasurable without orgasm, but only to differentiate the appetitive pleasure driven by the neural mechanisms of arousal and desire from the consummatory pleasure and satisfaction driven by orgasm (see Figure 9).

The ability to “let go” into an orgasm comes with sexual experience for both young men and women [243,244]. In a study of copulatory debut that divided participants at age 16 into “early” vs. “late” debut groups, significantly more females and males had their first coital experience with an “engaged partner” or a “steady boyfriend/girlfriend” in the late group compared to the early group [245]. Although at a superficial level this raises the question of what comes first, orgasm or partnering, it would seem that the two are driven by sexual arousal and desire, with orgasm perhaps acting to “seal the deal”, if what studies in rats have suggested at a mechanistic level has any bearing on primary experiences with sexual pleasure in humans. It is also the case that orgasmic pleasure is not stationary, but rather changes over the lifespan with more and more sexual experience. The sensory and emotional attributes that constitute a “peak” sexual experience is likely to change throughout the lifespan (e.g., [246]), but may well have qualitative descriptions similar to “…those moments of deep connection in which both lovers are psychologically and sexually accessible, engaged and responsive to whatever lies deep within.” [247]. This is the kind of connection alluded to by Stendahl in 1822 and cited explicitly by Krafft-Ebing in 1886. Figuratively speaking, it is the “path of Cupid’s Arrow”, the kind of connection that likely reflects the ability of opioid reward to sensitize DA and OXT systems in sexual circumstances so that they act on one another to facilitate sexual arousal and desire, pleasure, and bonding, perhaps in that order, so that these qualitative and transformational experiences with sexual pleasure can occur. Sexual pleasure feeds forward to augment sexual motivation and desire [248] and likely extends to person-related cues (e.g., shape of face, eyes, hair, body type, ethnic characteristics, interpersonal attributes), places, or even objects, conditioned by Pavlovian associations, and/or particular appetitive and consummatory sexual behaviors (e.g., body movements, use of toys, etc.) conditioned by response-reinforcer (operant) associations with arousal and pleasure [249]. Together, Pavlovian and operant associations with sexual arousal and pleasure partially define a preference for a familiar “type” and the movements that become one’s typical pattern of sexual response, including particular paraphilias [19,23].

## 7. Conclusions

It is clear that sexual experience generates plasticity in the brain that allows sexual reward to modulate the way predictive cues come to focus attention, augment sexual arousal and desire, and facilitate bonding. These are critical elements for the creation of partner preference and mate choice, especially for familiar partner-related features. The action of opioids as substrates of sexual reward facilitates the cascade of molecular events that lead to these enhanced responses, essentially the “personification of reward” that allows animals to differentiate the features they find attractive and prefer in potential sex partners. In Figure 8, we propose a process in which opioids such as β-endorphin derived from cell bodies in the arcuate nucleus bind to μ opioid receptors and produce the sensations of reward and pleasure that grow during sexual interaction, reaching their peak at orgasm. Through second-messenger and molecular actions, opioids sensitize DA, AVP, and OXT systems in hypothalamic and limbic structures to promote attention, arousal, and bonding with cues that predict sexual reward. These processes are then summed in the cortex and translate to a conscious awareness of attraction and desire.

Although partner features that depict health and the propensity for protection and survival of offspring may well be laid down genetically [250,251] and induce a preference for optimal inbreeding and outbreeding (e.g., [252,253]), such preferences rely on sexual imprinting, a phenomenon that is supposed to enable adults to recognize members of their own species for reproduction and parenting [254]. However, cross-fostering studies between goats and sheep indicate that experience with maternal care determines the species chosen for first sexual experiences, regardless of whether the species is one’s own [49]. Additionally, presumably the animals still have to copulate to seal the deal. Indeed, Fillion and Blass [255] found that male rats reared with dams that had their teats painted daily with citral odor investigated, copulated, and ejaculated more readily with citral-scented receptive females compared to unscented females, suggesting that the odor was paired with reward through suckling and feeding. Subsequently, Ménard et al. [58] examined whether a similar lemon odor paired with tactile reward in neonatal male rats would alter the subsequent expression of CEP. Newborn Long-Evans male rats were separated from their mothers each day beginning on Postnatal Day 1 and placed into a Plexiglas cage that contained either unscented or citral-scented bedding. During each trial, rats were stroked from head to toe with a soft, narrow paintbrush, after which they were returned to their mothers. When the males were adults, they were given their first sexual experience in an open field with two females, one scented with lemon and the other unscented. Males in the paired group copulated and ejaculated preferentially with the lemon-scented female whereas males in the unpaired group showed no preference. Pairing a neutral odor with a reward state in infancy generates a preference in male rats to ejaculate with sexually receptive females bearing the same odor in adulthood. As Lorenz noted in his seminal paper, the first experience of a greylag goose seeing a large object immediately after hatching induced a process of imprinting that compelled the gosling to follow Lorenz as if he was the gosling’s mother.

It is clear that first experiences with sexual arousal and reward induce sexual imprinting through a process of plasticity in the brain that allows sexual reward to modulate the way predictive cues come to focus attention, augment sexual arousal and desire, and facilitate bonding. These are critical elements for the creation of partner preference and mate choice. The actions of opioids as substrates of sexual reward facilitate the cascade of molecular events that lead to these enhanced responses, essentially the “personification of reward” that allows animals to discriminate and attend to the familiar features they find attractive and prefer in potential sex partners or objects associated with sexual reward. Obvious ethical constraints make it virtually impossible to manipulate, much less test, conditioning during critical first experiences with sexual reward that can occur any time across the lifespan in humans. We note that sexual interests span a continuum of preferences, from particular features and specific behavioral patterns to fetishes and paraphilias, as well as sexual orientation based on anatomical sex and perceived gender [19,23]. The wide variety of idiosyncratic features that can be conditioned in our species means that everyone will be found desirable by at least another, assuming that they possess the ability for competent sexual stimulation. Additionally, once conditioned, the type is not easy to extinguish, although it is likely iterative with new sexual “firsts” that alter the perception and definition of sexual reward into the realm of peak experiences.

Perhaps all species on this planet have sex for pleasure most of the time. Mating can occur—or not—with a bit of luck in a mating context, but mating is not critical for the individual so long as enough individuals in each generation of a species are successful at it. Attraction to particular individuals works hand-in-hand with the desire for sex. Such sexual bonding, even for a short time, requires conditioning and expectation of reward to produce effective appetitive responses toward preferred partners. First experiences with sexual reward matter in this, as they set the stage for the maintenance of variability that is the hallmark of hybrid vigor and continued survival within a species.

## Figures and Tables

**Figure 1 ijms-23-08928-f001:**
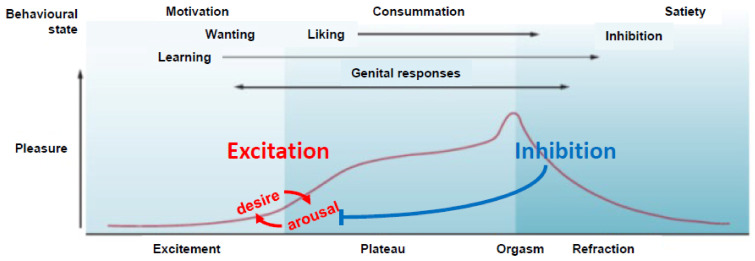
Depiction of Masters’ and Johnson’s “EPOR” model of human sexual response and related the activation of corresponding excitatory (red) and inhibitory (blue) brain regions. Opioid and serotonergic transmissions at orgasm activate inhibitory feedback on mechanisms of sexual arousal and desire. Adapted with permission from Georgiadis, Kringelbach, and Pfaus [14] (2012. Nature Publishing Group).

**Figure 2 ijms-23-08928-f002:**
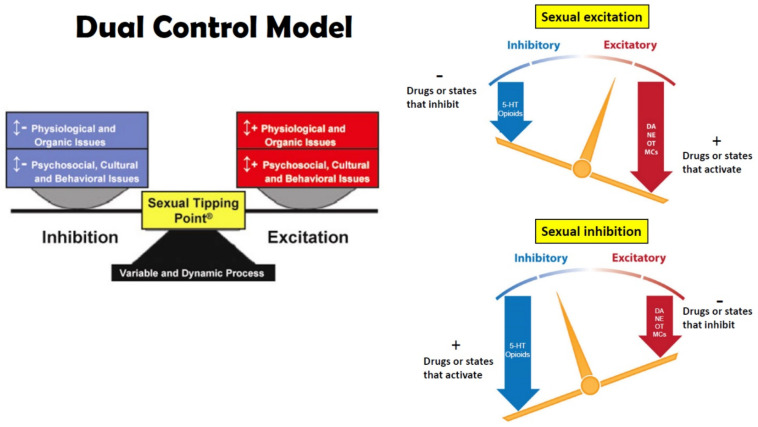
Left: dual control model of Perelman [1] (Reprinted with permission from Perelman, 2006, Wiley) that depicts sexual excitation and inhibition forming around a labile sexual tipping point. The dual-control model is based on work by Bancroft and Janssen [4], who based their model on the work of Gray [24] for fear conditioning. Gray, in turn, based his ideas on those of Pavlov [25] for the role of excitation and inhibition of cortical function in conditioned reflexes. Right: neurochemical mechanisms and processes associated with sexual excitation and inhibition. DA = dopamine, NE = norepinephrine, OT = oxytocin, MC = melanocortins, 5-HT = serotonin, reprinted with permission from Pfaus [3] (2009, Elsevier).

**Figure 3 ijms-23-08928-f003:**
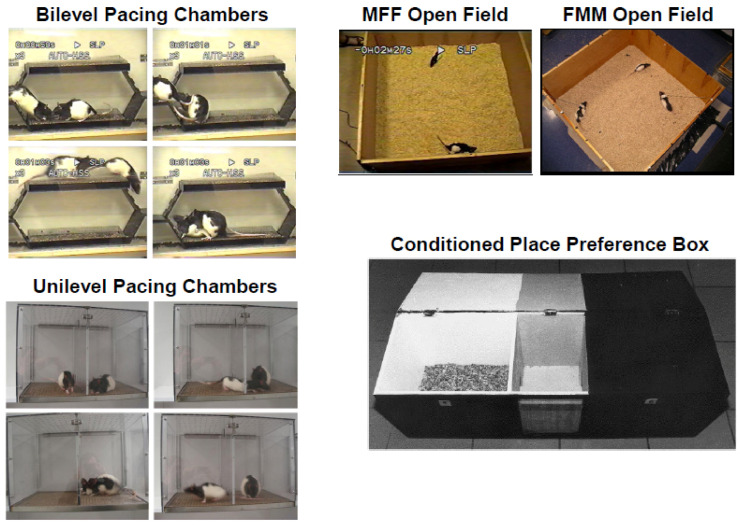
**Left**: Bilevel and unilevel pacing chambers used for conditioning of partner preferences (adapted from Pfaus et al., [53] with permission. 2015, Elsevier). Pacing chambers allow the female to regulate copulatory contact with the male by either running from level to level (**top**) or by darting through small holes in a divider that are big enough to allow the female to pass through, but too small to allow the male to pass through (**bottom**). In the bilevel pacing chambers, females make a head-wise orientation to the male (**upper left panel**), then hop over him exposing their anogenital area (**top right panel**). Females then run away, forcing the male to chase them (**bottom left panel**), after which they run to the original level and assume a lordosis crouch, allowing the male to mount with intromission (**bottom right panel**). Similarly, in unilevel pacing chambers females can pass through the hole into the male’s side (**top left** and **right panels**) and solicit an intromission (**bottom left panel**), then exit the male’s side (**bottom right panel**). Each of these sequences depict the initiation and termination of a copulatory bout by the female. **Right top**: Open fields used for testing conditioned partner preferences and CEP (adapted from Pfaus et al., [53] with permission. 2015, Elsevier). To test CEP, a male is placed into the open field with two females, one scented or jacketed and the other unscented or unjacketed. The male can then copulate freely with either one. To test partner preferences in females, a female is placed with two tethered males, one scented or jacketed, and the other unscented or unjacketed. Males are tethered to avoid their tendency to huddle together. Females compete, so they can remain untethered. **Right bottom**: Apparatus used to condition and test place preferences after sexual reward training (Reprinted with permission from Paredes and Vasques [54]. 1999. Elsevier). After a requisite amount of copulatory stimulation, a female or male is placed on one side of the box (usually the light side as this is typically not their naturally preferred side). This is contrasted without stimulation, after which the female or male is placed into the naturally preferred (e.g., dark) side. If the copulatory stimulation induces a reward state, then on a final test, after the animal is placed into the middle compartment, it will move to the side with cues that predict the reward state and spend more time there compared to the other side.

**Figure 4 ijms-23-08928-f004:**
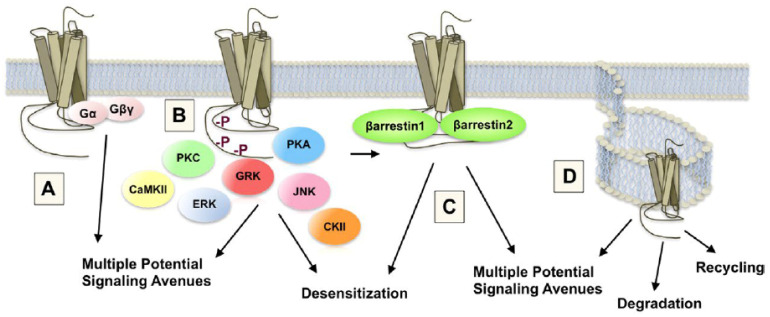
Structure and function of the μ opioid receptor and interaction with second messenger systems. The schematic demonstrates key points in opioid receptor signaling and regulation that was shown to be influenced by differential agonist occupation. **A**, heterotrimeric G proteins represent 16 individual gene products for Gα, 5 individual gene products for Gβ and 11 for Gγ proteins. Together, the diversity arising from heterotrimeric G protein subunit composition presents a gateway to potentially high diversification of agonist-directed coupling between μ opioid receptor and G proteins. These interactions can determine access to secondary cascade activation. **B**, the μ opioid receptor can be phosphorylated in response to agonist occupation by multiple kinases, each of which has multiple isoforms. Phosphorylation by a particular kinase may dictate secondary cascade interactions or subsequent receptor fate. CKII, casein kinase II. **C**, receptor interaction with scaffolding partners such as β-arrestins can be dependent or independent of receptor phosphorylation. Agonist occupancy may determine these interactions with potential binding partners. Such interactions can prevent (desensitization) or promote subsequent signaling. **D**, the μ opioid receptor can be internalized in response to agonist occupancy. Endocytosis may involve clathrin- or caveolin-dependent processes and may result in the activation of subsequent signaling pathways, receptor recycling or degradation. Reprinted with permission from Raehal et al. [145] (2011, The American Society for Pharmacology and Experimental Therapeutics).

**Figure 5 ijms-23-08928-f005:**
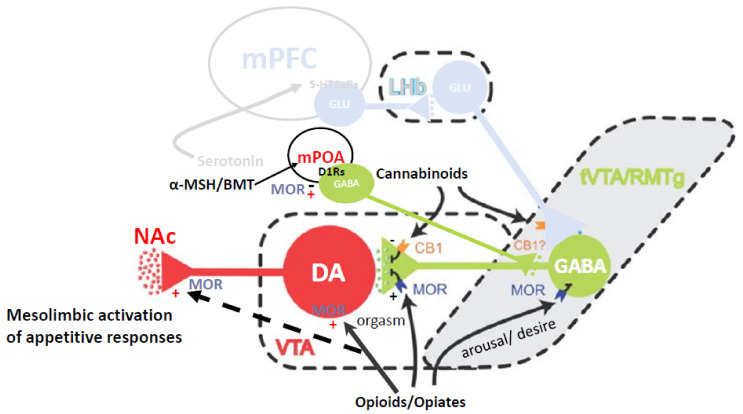
Neuroanatomical mechanisms of opioid inhibition and sensitization of mesolimbic and incerto-hypothalamic DA transmission. Opioids such as β-endorphin disinhibit dopamine release during sexual arousal and desire, but inhibit it at orgasm. This inhibition, however, results in sensitization of dopamine in subsequent trials [163,164]. Adapted from Barrot et al. [165] (2012, Barrot et al.)

**Figure 6 ijms-23-08928-f006:**
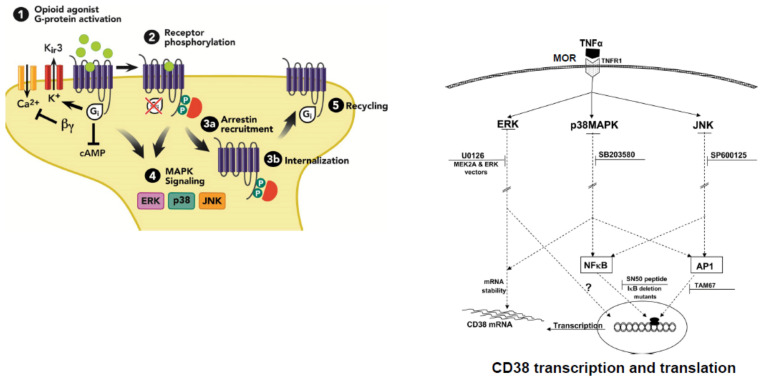
Stimulation of MAP kinase signaling by opioid agonist actions on μ opioid receptors leads to a cascade of second messenger related events (**left**), ultimately amplifying the transcription and translation of CD38 (**right**), a protein critical for Ca^2+^ mobilization within OXT neurons and augmentation of OXT neurotransmission when stimulated.

**Figure 7 ijms-23-08928-f007:**
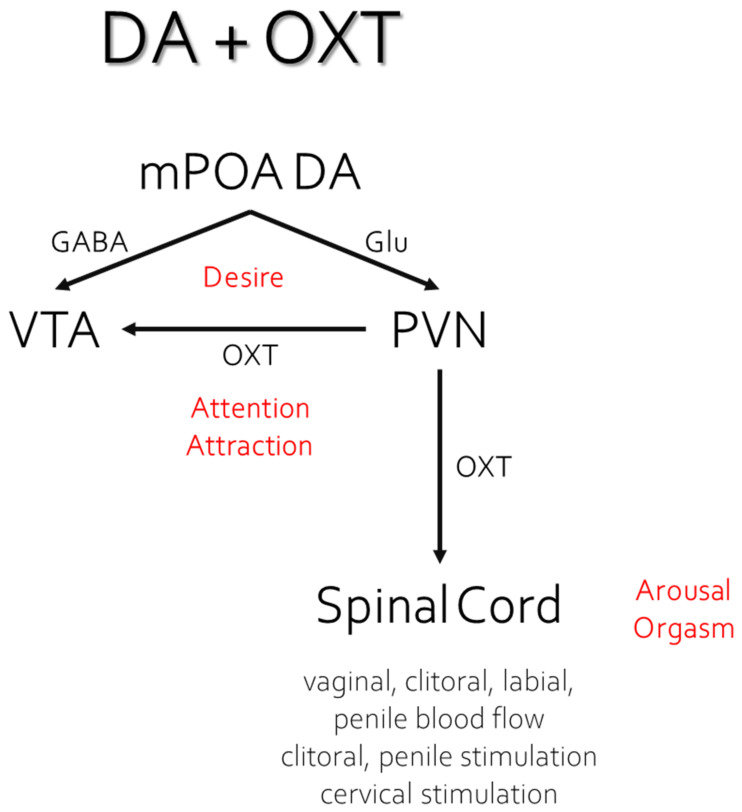
Coordination of genital blood flow, sexual desire, attraction, and orgasm by the interaction of DA and OXT in response to conditioned sensory cues; based on Melis and Argiolas [196]. Details are found in the text.

**Figure 8 ijms-23-08928-f008:**
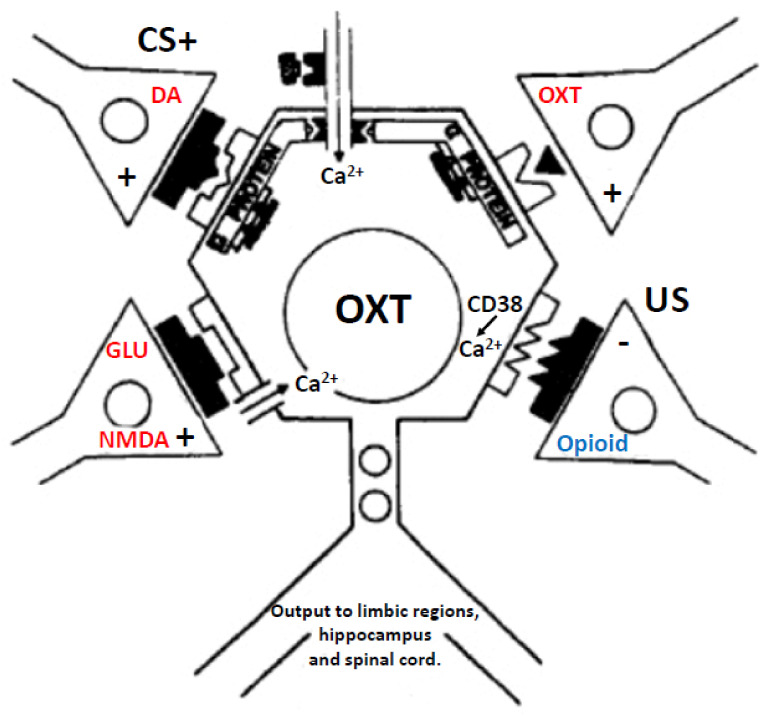
Synaptic inputs to OXT neurons in the PVN and possibly the SON. In this scheme, the CS + signal is carried by augmented incerto-hypothalamic DA transmission driven by sensitized mesolimbic DA. The US signal is carried by opioid transmission acting on μ opioid receptors. Although this signal inhibits OXT cell membranes, it leads to an augmentation of CD38 gene transcription, mobilization of Ca^2+^, and sensitization of OXT transmission when neurons are stimulated (in this case by DA, but potentially also by OXT and N-methyl D-aspartate [NMDA] glutamate [GLU] inputs).

**Figure 9 ijms-23-08928-f009:**
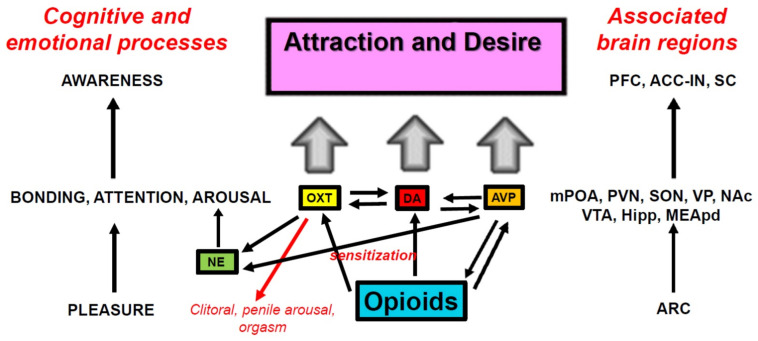
Mechanisms by which sexual pleasure and reward activate bonding, attention, and arousal to create motivation (desire) and attraction. Left: cognitive and emotional processes. Right: associated brain regions. ACC-IN = anterior cingulate-insula complex; ARC = arcuate nucleus; Hipp = hippocampus; NAc = nucleus accumbens (ventral striatum); MEApd = medial amygdala, posterior-dorsal region; mPOA = medial preoptic area; PFC = prefrontal cortex; PVN = paraventricular nucleus of the hypothalamus; SC = somatosensory cortex; SON = supraoptic nucleus of the hypothalamus; VP = ventral pallidum; VTA = ventral tegmental area.

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
