# Peer review of "Behavioral, Neural, and Molecular Mechanisms of Conditioned Mate Preference: The Role of Opioids and First Experiences of Sexual Reward"

_ijms, 2022, doi:10.3390/ijms23168928_

Round 1
Reviewer 1 Report
This is an exceptionally well-written, comprehensive and timely review and I really enjoyed reading it. I only have a few comments, which I am confident the authors will be able to address.
The authors speak about sexual behavior and underlying molecular mechanisms that drive motivation and arousal in humans and rodents. However, the current Figure 1 only highlights brain areas involved in sexual behavior in humans. Is it feasible to recreate this Figure for rats/mice, at least in part? Alternatively, it would be nice to summarise existing evidence about the respective brain regions in rats/mice in tabular form.
Page 4: The authors mention fluoxetine and paroxetine as examples of SSRIs and write: 'Facilitation of serotonin turnover...'. However, SSRIs inhibit reuptake of serotonin from the synaptic cleft, so in that sense, they inhibit serotonin turnover, rather than facilitating it. I think the sentence needs to be adjusted accordingly.
Figure 3: What source are the shown pictures taken from? A bit more information in the Figure legend about the design of the boxes, the testes behaviours and the idea behind the paradigms would be appreciated. Also, please indicate what each image precisely shows (initiation of mating, copulation etc.).
Please be consistent with abbreviations for oxytocin and vasopressin throughout the manuscript: OT/OXT and AVP/VP. The same goes for OT/OXT receptors.
Page 18: Some of the statements made here about projections of parvocellular and magnocellular oxytocin neurons are either inaccurate or oversimplified. In fact, only very few parvocellular oxytocin projections to non-hindbrain areas in rodents have been reported to this date (Althammer et al., 2017, Journal of Neuroendocrinology). The vast majority of parvocellular OT neurons in rats and mice project to the brainstem, PAG or spinal cord (hindbrain read). Also, direct contact from parvocellular PVN OT neurons to magnocellular OT neurons in the SON, has been reported (Eliava et al., 2016, Neuron), whereby parvocellular OT neurons coordinate release of OT from magnocellular SON OT neurons into the bloodstream. Magnocellular neurons in contrast, have been demonstrated to project to more than 50 forebrain regions (Knobloch et al., 2012, Neuron) so they do not 'just project' to the pituitary for release of neuropeptides into peripheral circulation. In addition, the authors should be careful with mixing AVP and OT projections and terminology within the same sentence. Parvocellular OT neurons are distinct from preautonomic/parvocellular AVP neurons within the PVN for instance, and projection patterns of AVP and OT neurons are vastly different for the respective cell types.
I am unsure how helpful Figure 7 really is - there are too many brain regions, arrows, interactions, transmitters and question marks. I suggest to either remove it or recreate it in a much simpler fashion.
Page 21: '..if what our rats'. Not sure the term 'our rats' is appropriate, I suggest to go with something a bit more neutral, along the lines of: '..studies in rats have shown/suggest'.
Author Response
Reviewer 1
This is an exceptionally well-written, comprehensive and timely review and I really enjoyed reading it. I only have a few comments, which I am confident the authors will be able to address.
The authors speak about sexual behavior and underlying molecular mechanisms that drive motivation and arousal in humans and rodents. However, the current Figure 1 only highlights brain areas involved in sexual behavior in humans. Is it feasible to recreate this Figure for rats/mice, at least in part? Alternatively, it would be nice to summarise existing evidence about the respective brain regions in rats/mice in tabular form.
Response: We removed the brain activation part of the figure as we were only referring to the 4-stage model and the theoretical mechanisms that underlie it, not the regional brain activation that accompanies each stage. Part of the problem in relating human fMRI or PET studies as depicted in that figure with rodent studies that are derived largely from Fos, other immediate-early gene activation, or 2-DG studies as markers of neuronal activation, is that the animal people largely study subcortical and hypothalamic regions whereas the human people study cortical and some limbic regions. The spatial resolution of fMRI or PET really cannot distinguish substructures of the human hypothalamus, whereas the Fos activation in the rat or mouse cortex is often overlooked. Indeed, this is a very important issue and it really requires a paper in its own right.
Page 4: The authors mention fluoxetine and paroxetine as examples of SSRIs and write: 'Facilitation of serotonin turnover...'. However, SSRIs inhibit reuptake of serotonin from the synaptic cleft, so in that sense, they inhibit serotonin turnover, rather than facilitating it. I think the sentence needs to be adjusted accordingly.
Response: This has now been changed on P. 6 to “Blocking serotonin reuptake…”
Figure 3: What source are the shown pictures taken from? A bit more information in the Figure legend about the design of the boxes, the testes behaviours and the idea behind the paradigms would be appreciated. Also, please indicate what each image precisely shows (initiation of mating, copulation etc.).
Response: More explanation and citations have been added to the figure caption.
Please be consistent with abbreviations for oxytocin and vasopressin throughout the manuscript: OT/OXT and AVP/VP. The same goes for OT/OXT receptors.
Response: This has been taken care of. It is OXT throughout.
Page 18: Some of the statements made here about projections of parvocellular and magnocellular oxytocin neurons are either inaccurate or oversimplified. In fact, only very few parvocellular oxytocin projections to non-hindbrain areas in rodents have been reported to this date (Althammer et al., 2017, Journal of Neuroendocrinology). The vast majority of parvocellular OT neurons in rats and mice project to the brainstem, PAG or spinal cord (hindbrain read). Also, direct contact from parvocellular PVN OT neurons to magnocellular OT neurons in the SON, has been reported (Eliava et al., 2016, Neuron), whereby parvocellular OT neurons coordinate release of OT from magnocellular SON OT neurons into the bloodstream. Magnocellular neurons in contrast, have been demonstrated to project to more than 50 forebrain regions (Knobloch et al., 2012, Neuron) so they do not 'just project' to the pituitary for release of neuropeptides into peripheral circulation. In addition, the authors should be careful with mixing AVP and OT projections and terminology within the same sentence. Parvocellular OT neurons are distinct from preautonomic/parvocellular AVP neurons within the PVN for instance, and projection patterns of AVP and OT neurons are vastly different for the respective cell types.
Response: This paragraph has been rewritten extensively in accordance with the suggestions of the reviewer (P. 29-30).
I am unsure how helpful Figure 7 really is - there are too many brain regions, arrows, interactions, transmitters and question marks. I suggest to either remove it or recreate it in a much simpler fashion.
Response: The figure has been simplified.
Page 21: '..if what our rats'. Not sure the term 'our rats' is appropriate, I suggest to go with something a bit more neutral, along the lines of: '..studies in rats have shown/suggest'.
Response: This now reads, “…if what studies in rats have suggested at a mechanistic level…”

Reviewer 2 Report
A very interesting and well written review of the mechanisms shaping sexual preference based on experience and reward. The description of experiments and results uses a very good narrative style. The figures are well composed and drawn. I have some minor comments, but otherwise happy to recommend for publication. Figure 2, the arrow labels on the right hand side are too small to be clearly read. The layout is good otherwise and this is likely partly due to poor image quality in the review copy, but please make sure a vector (such as EPS) rather than bitmap based version is used. Page 5, mid section 2 introduction, odd that line starts with 'However' since it appears to support previous result. Is there maybe a 'not' missing from the sentence? Page 9, CLS and VCS are used without being defined Page 11, section 2.4, it would be useful to say here if the OXT and AVP injections were peripheral or central (ICV) Page 11, 'assess' I think should be 'access' Page 11, the action of 'oryzon' here needs some more explanation Page 13, the definition of DALA comes at its second use Figure 4 legend, extra full stop at the end Page 14, the final sentence in its use of 'toward' is quite ambiguous, is the preference toward reduced? Page 16, 'showing that opioids acting' should be 'showing that opioids acting on' Page 18, this section should also consider the central actions of magnocellular neurons acting via dendritic release Page 21, lower paragraph, the sentence about dividing based on age into debut groups is not very clear
Author Response
Response to reviewers:
Reviewer 2
A very interesting and well written review of the mechanisms shaping sexual preference based on experience and reward. The description of experiments and results uses a very good narrative style. The figures are well composed and drawn. I have some minor comments, but otherwise happy to recommend for publication.
Figure 2, the arrow labels on the right hand side are too small to be clearly read. The layout is good otherwise and this is likely partly due to poor image quality in the review copy, but please make sure a vector (such as EPS) rather than bitmap based version is used.
Response: The labels have been made larger and more legible.
Page 5, mid section 2 introduction, odd that line starts with 'However' since it appears to support previous result. Is there maybe a 'not' missing from the sentence?
Response: To make this more clear, the sentence has been changed to “Despite species’ differences in copulatory behavior, it is very likely that similar, if not identical, mechanisms underlie those experiences in humans…”
Page 9, CLS and VCS are used without being defined
Response: These abbreviations are now defined here.
Page 11, section 2.4, it would be useful to say here if the OXT and AVP injections were peripheral or central (ICV) Page 11, 'assess' I think should be 'access' Page 11, the action of 'oryzon' here needs some more explanation
Response: We now state that the injections were subcutaneous. Assess was changed to access. And we have provided a sentence with more explanation of oryzon’s action.
Page 13, the definition of DALA comes at its second use Figure 4 legend, extra full stop at the end
Response: Definition of DALA is now made at its first use. Extra full stop has been taken out.
Page 14, the final sentence in its use of 'toward' is quite ambiguous, is the preference toward reduced?
Response: This has been clarified by stating that they display a preference for the familiar female.
Page 16, 'showing that opioids acting' should be 'showing that opioids acting on'
Response: This has been changed accordingly.
Page 18, this section should also consider the central actions of magnocellular neurons acting via dendritic release
Response: We have rewritten this section to take into consideration both reviewers’ suggestions.
Page 21, lower paragraph, the sentence about dividing based on age into debut groups is not very clear
Response: This sentence has been changed to, “The sensory and emotional attributes that constitute a “peak” sexual experience is likely to change throughout the lifespan” to make it clearer.
